# Differentiating Dyes: A Spectroscopic Investigation into the Composition of Scarlet Bloodroot (*Haemodorum coccineum* R.Br.) Rhizome

**DOI:** 10.3390/molecules28217422

**Published:** 2023-11-03

**Authors:** Matheus Carpinelli de Jesus, Taylah Church, Johanna A. Wapling, Raelene Collins, Gregory J. Leach, David Leach, James J. De Voss, Joanne T. Blanchfield

**Affiliations:** 1School of Chemistry and Molecular Biosciences, The University of Queensland, Brisbane, QLD 4072, Australia; j.devoss@uq.edu.au; 2Menzies School of Health Research, Charles Darwin University, Darwin, NT 0810, Australia; taylah.church@menzies.edu.au (T.C.); jo.wapling@gmail.com (J.A.W.); raelene.collins@menzies.edu.au (R.C.); gregj.leach@gmail.com (G.J.L.); 3Research and Development, Integria (MediHerb), Brisbane, QLD 4113, Australia; david.leach@integria.com

**Keywords:** *Haemodorum coccineum*, natural products, phenylphenalenones, traditional medicine, xanthanes

## Abstract

*Haemodorum coccineum*, commonly known as scarlet bloodroot, is a plant native to New Guinea and the northern most parts of Australia. The highly coloured *H. coccineum* is used by communities in Larrakia country for dyeing garments and occasionally to treat snake bites. Previous studies into *H. coccineum* have focused on its taxonomic classification, with this being the first evaluation of the chemical composition of the plant. *Haemodoraceae* plants are reported to contain phenylphenalenones (PhPs), which are highly conjugated polycyclic oxygenated aromatic hydrocarbons. We report the characterisation of 20 compounds extracted from the rhizome of *H. coccineum*: four sugars and 16 compounds belonging to the PhP family. The compounds include five aglycones and seven glycosylated compounds, of which four contain malonate esters in their structures. Characterisation of these compounds was achieved through 1D and 2D NMR, MS analysis and comparison to the known phytochemistry of other species from the *Haemodorum* genus. Preliminary anti-microbial activity of the crude extract shows significant inhibition of the growth of both gram-positive and gram-negative bacteria, but no activity against *Candida albicans*.

## 1. Introduction

*Haemodorum coccineum* (also known as scarlet bloodroot) belongs to the *Haemodoraceae* family which is confined to the southern hemisphere [1]. Current studies indicate that the phytochemical composition of the *Haemodorum* species is characterised by the abundance of phenylphenalenones (PhPs) [2,3,4,5,6,7,8]. This is a property shared with other genera in the *Haemodoraceae* family. These highly conjugated PhPs are responsible for the characteristic deep red colouration of members of the *Haemodoraceae* family [1].

*H. coccineum* is a plant of great economic importance to some Aboriginal and Torres Strait islander (ATSI) communities. The plant is used as a dyestuff in the production of tradeable/wearable goods. A pharmacopeia from 1988 reports a personal communication indicating the traditional medicinal use of the rhizome in the treatment of snake bites [9]. A subsequent pharmacopeia, published in 1993 [10], includes reference to its use as a topical wash for skin lesions. This was a liquid decoction from crushed root-stock pieces.

Reported here is the first study into the phytochemical composition of *H. coccineum* R.Br. A study in 1955 into the composition of *H. corymbosum* Vahl. misassigned it as a taxonomic synonym to *H. coccineum* [1]. The taxonomic clarification was only published in 1987 [11]. In the analysis of *H. corymbosum* extracts, one compound was isolated and synthetic derivatives produced. The isolated compound was haemocorin, a 9-phenylphenalen-1-one glycoside. The glycoside consisted of a lactose unit—two *β*-D-glucose units with a 1–4 linkage. This type of compound is common in plants belonging to the *Haemodoracae* family.

The study described herein, aims to characterise the composition of *H. coccineum* rhizome, as this is the plant part used traditionally. In this endeavour, a combination of sonication-assisted extraction and pressurised hot water extraction (PHWE) [12] were used to generate crude extracts. Organic and aqueous partitions were then progressively fractionated using solid-phase extraction, flash column chromatography and HPLC to isolate the compounds described.

## 2. Results and Discussion

### 2.1. Extration and Generation of Extract

Plants belonging to the *Haemodoraceae* family have previously been reported to contain highly conjugated polycyclic oxygenated aromatic hydrocarbons. To obtain a comprehensive profile of the PhPs present in the rhizome of *H. coccineum,* sonication of a suspension of the ground material in methanol was conducted. Small aliquots of this dense, red extract were analysed using LC-PDA-MS and 1D NMR techniques. This initial screening allowed for some structural information to be acquired to inform the techniques employed for the isolation and characterisation of the secondary metabolites present in *H. coccineum*. 

### 2.2. Fractionation of Methanolic Extract

With the dense, red extract in hand, evaluation of the phytochemical composition of *H. coccineum* started with the addition of water to the methanolic decoction and extraction with chloroform. The aqueous methanol layer that remained was divided into two portions for separate analysis. The first was fractionated using a C18 solid-phase extraction (SPE) cartridge, resulting in the characterisation of six compounds: sucrose (**1**), α-D-glucose (**2**), β-D-glucose (**3**), 6″-*O*-malonyldilatrin (**4**), **5** and **6**. Reverse-phase high-performance liquid chromatography (RP HPLC) was required to isolate lachnanthocarpone dimethylether (**7**) and haemodordioxolane (**8**). The other part of the aqueous methanol was fractionated directly using RP HPLC, yielding compounds **1**, **3**, **6**, **10**, dilatrin (**12**) and fructose (**20**). The chloroform layer was fractionated on a normal-phase silica column. The nine compound-containing fractions obtained were pooled to yield compounds **4**, **16**, haemodordiol (**17**), **18** and **19** (Figure 1).

Of the compounds characterised, only five were obtained as a pure compound (**2**, **4**, **10**, **12** and **19**). All other compounds were characterised from binary mixtures. The fact that many of these are known compounds with reported chemical shifts made it possible to discern characteristic peaks and correlations in the 2D NMR spectra and elucidate their structures.

Compounds **1**–**3** were identified as sucrose (**1**), α-D-glucose (**2**), and β-D-glucose (**3**), using ^1^H NMR and COSY experiments. The NMR data was consistent with the literature [13].

### 2.3. Malonate or Allophanyl?

Compound **4** was isolated both from the fraction that was eluted in 30% methanol from the C-18 SPE cartridge and from the flash column chromatography of the chloroform soluble partition. Elucidation of the structure of compound **4** was crucial in enabling the identification of the other compounds contained in the plant. Analysis of the LC-PDA-MS data revealed that the compound exhibited absorption peaks at λ_max_ 470, 374 and 274 nm, and possessed molecular ion peaks in both positive (*m*/*z* 567 [M + H]^+^) and negative ion modes (*m*/*z* 565 [M − H]^−^, *m*/*z* 1131 [2M − H]^−^). This MS data suggested a molecular mass of 566 gmol^−1^. MS data also showed ions at *m*/*z* 319 [M + H − 86 − 162]^+^ and *m*/*z* 317 [M − H − 86 − 162]^−^, corresponding to the phenylphenalenone core upon the loss of a hexose (-162 Da) and an 86 Da moiety. The loss of 86 Da observed in positive mode, *m*/*z* 481 [M − 86]^+^, was hypothesised to be the loss of a malonate (COCH_2_COOH) or a allophanyl group (CONH_2_CONH_2_); both groups have been reported as features of compounds in the *Haemodoraceae* family [14,15,16].

The region of the ^1^H NMR spectrum between δ_H_ 4.66 and 2.37 indicated the presence of a number of CHO resonances consistent with a sugar unit in the structure. COSY was used to connect the signals to demonstrate that they corresponded to a hexose unit, starting from a δ_H_ 4.66 anomeric doublet. HSQC aided with the assignment of the carbon signals observed for the hexose unit, allowing for an assignment consistent with a glucose residue. The anomeric proton coupling of 7.8 Hz to H-2 suggested this was a ꞵ-glucose. Next the assignment of the signals in the aromatic region was undertaken. The signals at δ_H_ 7.22, 7.35 and 7.48 showed a large vicinal coupling (*J* = 7.5 Hz) and small long-range couplings (*J* = 1.5, 1.2 Hz) to each other, allowing for the assignment of ring D (Figure 2). Only a limited number of compounds have been isolated from *Haemodorum* plants, where a malonate/allophanyl, a sugar unit and a phenyl group are present. This suggested that the compounds were likely to be phenylphenalenones or benzo[*de*]isochromenones. The distinguishing features between these two cores are the lactone ring A in benzo[*de*]isochromenones versus the α,ꞵ-unsaturated cyclohexanone ring A in phenylphenalenones (Figure 2). The proton signal at δ_H_ 7.15 integrated to one proton, and no signal was observed at the δ_H_ 5.80–5.70 region where a methylene of a lactone would be expected to resonate (as seen subsequently in compounds **6**, **9**, **10** and **19**) [14,15]. Similarly, the presence of a methyl ether signal at δ_H_ 3.94 (δ_C_ 54.1) correlating to a quaternary carbon signal at δ_C_ 154.0 in the HMBC spectrum, supported the assignment of the core as a phenyphenalenone with a methyl ether at C2, rather than an isochromenone. At this stage, it was possible to assign the compound as 6-*O*-β-D-glucopyranosyl-5-hydroxy-2-methoxy-7-phenylphenalen-1-one, with H6″ of the glucose correlated to C1‴ (δ_C_ 168.5) of either a malonyl or an allophanyl ester. The data for this compound matched exactly those reported for a 6″-*O*-allophanyl ester **4′** in 2002, in a study of *Xiphidium caeruleum* [16], but also matched those for the 6″-O-malonyl ester **4** reported in a 2012 study of *Wachendorfia thyrsiflora* L. [15]; both publications originated from the same research group.

The unusual urea derivative postulated by Opitz et al. seemed an unlikely natural product structure [16]. A more commonly found group in natural products is the malonate group (HOCOCH_2_COOH), which has a similar mass to that of allophanic acid (Table 1).

We thus carefully compared the reported shifts of compounds **4′** and **4** with those of the compound isolated from *H. coccineum*. The ^13^C NMR chemical shifts assigned to the carbonyl groups of the substituent in each natural product were compared to chemical shifts reported for a synthetic malonyl methyl ester [17], and for a synthetic allophanyl methyl ester [18] (values provided on structures below). The differences in the reported shifts for compounds **4′** and those for the synthetic malonate and allophanyl esters are summarized in Table 2. The chemical shifts of carbonyl groups in the natural products are most closely aligned to the malonate methyl esters. Analysis of their ^13^C NMR data reported for **4′** revealed that the chemical shifts assigned to the allophanic acid carbonyl groups are conspicuously unusual for urea carbonyls, which might be expected to appear 15 to 20 ppm further upfield (Table 2).

Additional support for the assignment of the substituent as a malonate group was provided via the comparison of the calculated molecular weight for the two proposed derivatives with the accurate mass data reported for the natural products (Table 3). This also showed that the isolated compound is the malonyl and not the allophanyl derivative. In addition, it suggests that the original reports of the allophanyl-derived compounds as natural products are incorrect and should be revised to malonates [16,19].

Malonic acid already plays a critical role in the biosynthesis of these polyketide compounds, which further supports the revised malonate ester structures (Figure 2). The methylene protons of the malonyl ester were not observed in the ^1^H NMR, likely due to an exchange with deuterium from the CD_3_OD solvent. This signal was also not reported in the literature [16]. The complete characterisation of **4** was achieved using NMR from a >90% pure sample isolated with HPLC through the use of 2D NMR spectroscopy.

### 2.4. Two Series of Glycosilated Phenylphenalenone

With the spectroscopic data for **4** fully assigned, the identification of two series of related glycosylated PhP derivaties was achieved (compounds **4**, **5**, **6**, **9**, **10**, **11**, **12** and **19**). The first series derived from the same aglycone core, with variations on the methyl ether and mallonate groups (compounds **4**, **5**, **11** and **12**). The NMR spectra of **5** lacked the methyl ether signals (δ_H_ 3.94 and δ_C_ 54.1), while compound **12** possessed spectra analogous to **4,** but lacking the malonate-associated resonances with the expected molecular ion observed in positive mode at *m*/*z* 481.10 [M + H]^.+^, and negative mode at *m*/*z* 479.09 [M − H]^.−^.

The second series of PhP derivatives (**6**, **9**, **10** and **19**) consists of the 7-phenyl-benzo[*de*]isochromenone core with variations analogous to those seen in the first series, all with the diagnostic ~δ_H_ 5.7 signal corresponding to the H-3 protons. Compound **19** had a 6″-*O*-malonyl-glucose, based upon the appearance of the malonyl C2‴ (δ_C_ 40.6) and anomeric proton H-1″ (δ_H_ 4.61). Compound **9** was tentatively identified and only observed on the LC-PDA-MS, with positive (*m*/*z* 455.14) and negative ion (*m*/*z* 452.96) mode corresponding to **19,** but lacking the malonyl substituent. Compounds **6** and **10** presented a *para-*OH-substitution pattern on the phenyl ring (**10**, ~δ_H_ 7.34 (2′, 6′), 6.95 (3′, 5′)) of the benzo[*de*]isochromenone. They also displayed a 16 Da increase in their molecular ions compared to the analogous **19** and **9** which possess an unsubstituted phenyl D ring.

The structures for both **9** and **11** were proposed based on their UV profile and ESIMS data, and were informed by the other compounds isolated from the rhizome. Compound **9** was proposed as the biosynthetic precursor or hydrolysis artifact of compound **19**, while compound **11** was proposed to be the extraction artifact formed from the decarboxylation of the malonyl ester unit in compound **4**.

### 2.5. Pressurised Hot Water Extraction (PHWE)

Pressurised hot water extraction (PHWE) is a method adapted by A/Prof. Jason Smith in 2015 [12], which employs an espresso machine to extract plant material in an economically and environmentally friendly manner. Unlike sonication-assisted extractions, PHWE does not rupture the plant cells, producing an extract with less unwanted plant matter. Since only hot water is used in this extraction method, the compounds acquired would be most akin to boiling the plant, which is assumed to be the traditional method of preparing an extract. It was decided therefore, to use PHWE to extract a sample of *H. coccineum* rhizome for comparison with the more widely used methanolic extracts. PHWE produced a hot aqueous extract which was partitioned between chloroform and water. The organic partition was analysed using LC-PDA-MS and this indicated the presence of six compounds (lachnanthocarpone dimethylether (**7**), haemodordioxolane (**8**), compounds **9**–**11** and dilatrin (**12**)). Compounds **7**, **8**, **10** and **12** were isolated and characterised in higher quantities from the aqueous methanol partition, while compounds **9** and **11** were only detected with LCMS. The ^1^H NMR of compound **7** was unlike the other PhPs characterised thus far, due to the abundance of aromatic doublets. Each doublet integrated to one proton had coupling constants indicating adjacent protons; δ_H_ 8.61 and 7.55 (*J* = 8.4 Hz) and δ_H_ 7.59 and 6.88 (*J* = 8.0 Hz). The remaining aromatic signals (δ_H_ 7.42, 7.38, 7.36) integrated to five protons which, correlated to carbon signals (δ_C_ 128.7, 129.3, 127.7), were assigned to mono-substituted phenyl ring protons. Two signals at δ_H_ 4.07 and 3.84, integrating to three protons each and associated with carbon shifts of δ_C_ 55.2 and 55.7, were assigned as methyl ethers. HMBC showed a correlation between the carbonyl (C1) at δ_C_ 179.9 and OMe-6 (δ_H_ 3.84) and H-8 (δ_H_ 7.57). This arrangement indicated a 9-phenylphenalenone core with two methoxy substituents. The last aromatic signal was a singlet at δ_H_ 6.81, commonly assigned to C3 or C4 of a phenylphenalenone structure. The ^1^H and ^13^C NMR data reported by DellaGreca and colleagues indicate that these chemical shifts were congruent with lachnanthocarpone dimethylether (**7**).

Similarly, the ^1^H NMR for compound **8** had even fewer signals, indicating a highly substituted or more oxidised structure, and possessed a molecular mass of 318 gmol^−1^. Similar to previous structures, the aromatic signals associated with the phenyl group were characterised as a multiplet (δ_H_ 7.49–7.46) and a doublet (δ_H_ 7.42, 1.7 Hz) integrating for three and two protons, respectively. The protons at C9 and C8 were assigned as the doublets at δ_H_ 8.54 and 7.57, due to their downfield shift and coupling constant (7.5 Hz). A δ_H_ 6.09 signal, integrating to two protons, indicated a methylenedioxy acetal moiety; within a phenylphenalenone core, there are limited sites for such an acetal to form. Common positions for vicinal hydroxylation are C5 and C6, and this presents the required functionality for an acetal formation. The low molecular mass of 318 gmol^−1^ and the functionalities described leave limited possibilities. The NMR data for compound **8** were consistent with Urban’s haemodordioxolane, originally isolated from *Haemodorum simulans* [7].

### 2.6. Hydrolysis of PHW Extract

Since previous studies highlighted the abundance of PhPs in plants from the *Haemodoraceae* family, and PhPs **7**–**12** were present with a diversity of glycosylation and esterified substituents, hydrolysis was an important step to identify the full range of PhP aglycones present in this extract. The comprehensive understanding of the PhP core units present in the extract was key in confirming structural assignments. Hydrolysis of part of the PHW extract was achieved via heating the extract in methanolic HCl under reflux. The resulting solution was then extracted exhaustively with chloroform. The aqueous partition contained α-D-glucose (**2**) in abundance; the chloroform extract was fractionated over normal-phase silica, yielding compounds **13**–**18**. Compound **14** was analogous to **7** but lacked the methyl ether at C2 and may be a product of its (**7**) hydrolysis. Compound **15** was analogous to the aglycone of dilatrin (**12**) but with an additional methyl ether at δ_H_ 3.75, previously reported as xiphidone (**15**) [20]. The spectra of **16** were similar to **15** but with different substitution in the phenyl ring D. The splitting pattern indicated an *ortho*-substitution, leading to the suggestion of a 6-oxabenzo[*def*]crysenone core. NMR characterisation by Cooke and colleagues [21] of 2,5-dimethoxy-1*H*-naphtho [2,1,8-*mna*]xanthenone was consistent with **16**. (Note: the names 6-oxabenzo[*def*]crysenone and 1*H*-naphtho[2,1,8-*mna*]xanthenone refer to the same core (see Appendix A). In turn, the ^1^H NMR of **18** resembled that of **16** with a molecular mass difference of 14 Da; aa methyl ether signal (~δ_H_ 4.00) was also absent in the spectrum of **18**. The NMR data for **18** in CD_3_OD were consistent with DellaGreca and colleagues’ report [22], and those in CDCl_3_ were consistent with Brkljaca and colleagues [8]. The data for **13** were similar to those for **8** but lacking the methylenedioxy resonances; instead, a methyl ether (δ_H_ 3.68, δ_C_ 56.8) was observed at C6, and the NMR data were consistent with haemodorone (**13**), as reported by Dias and colleagues from *Haemodorum simplex* [6]. The overall signal distribution in the ^1^H NMR for compound **17** indicated another PhP structure with substantial substitution. Vicinal protons at H-9 (δ_H_ 8.61) and H-8 (δ_H_ 7.76) showed COSY correlations and a coupling constant of 8.9 Hz. HSQC reinforced these assignments, and their chemical shifts (δ_C_ 131.4, 146.9) were consistent with aromatic protons and conjugation with an ester. Haemodordiol, from *H. spicatium*, was found to exhibit an H-3 signal at around δ_H_ 6.53 with acetal-like character. The NMR reported by Brklijaka and colleagues [8] was crucial to the assignment of this compound. Compounds **17** and **16** were in a 1:6 (**17**:**16**) ratio in the ^1^H NMR, which meant the proton signals associated with phenyl ring D and H-4 were obscured. Based on the data acquired, it was possible to identify haemodordiol (**17**), as reported by Brklijaka and colleagues [8].

### 2.7. Preliminary Antimicrobial Activity

The crude methanolic extract of the rhizome was screened for antimicrobial activity against a panel of microbes including yeast, and gram-negative and gram-positive bacteria. The crude extract obtained from the rhizome was tested at a single concentration of 20 mg/mL, and demonstrated antimicrobial activity against *Staphylococcus aureus*, *Streptococcus pyogenes* and *Haemophilus influenzae* reference strains. Of note, partial inhibition (approximately 60%) was observed against *Pseudomonas aeruginosa* and *E. coli*, indicating the presence of compounds within the crude extract that could potentially inhibit these gram-negative bacteria at a higher concentration. Based on these initial data, antimicrobial testing of individual compounds with the determination of minimal inhibitory concentration is warranted; however, it would require larger amounts of raw product, which was beyond the scope of this study. No activity was observed when the extract was tested on the yeast strain *Candida albicans*. While phenylphenalenones are used by bananas (*Musa acuminata*) in its defence against fungal infection (*Mycosphaeralla fijienses*) [23,24], these or similar microbes were not tested in this study.

### 2.8. Rhizosheath Phytochemical Analysis

The rhizome sample was collected with substantial amounts of soil trapped within (rhizosheath). This soil was shaken loose and kept for analysis. The sample was loaded onto a soxhlet thimble and extracted with methanol. The extract was analysed using GCMS and LCMS. The LCMS analysis indicated the presence of compounds **8**, **13** and **15**, based on retention time, UV profile and mass spectrometric comparison; thus these compounds may serve some biological role within the soil. Three compounds were observed on the GCMS chromatogram of the extract, corresponding to three different compounds (glycerol, ethyl pyruvate and D-hexose).

## 3. Conclusions

Extraction of the rhizome of *H. coccineum* yielded 20 compounds (Figure 3); 16 secondary metabolites, with 14 PhPs fully characterised (**4**–**8**, **10**, **12**–**19**), two (**9**, **11**) being tentatively identified, along with compound **11** which is proposed to be an artifact of extraction. The abundance of PhPs in *H.* coccineum is consistent with the phytochemistry of other Haemodorum species. The remaining four compounds were primary metabolites, carbohydrates **1**–**3**, and **20**. Although the PhPs are previously characterised natural products, this is the first report of their presence in *H. coccineum* and the first report of the phytochemistry of this species. Some of the PhPs isolated possessed identical spectroscopic properties to previously reported allophanyl derivatives. It was found that these were in fact malonates that had later been reported from different sources. It was also observed that PhPs are responsible for the colouration of the rhizome.

The variety of aglycones isolated was greater than that seen in the glycosylated compounds. As most of these aglycones presented analogous functionality for glycosylation, it is possible there are more minor glycosylated compounds that remain to be detected.

The preliminary biological activity results, with significant activity against the gram positive, support the plant’s biological activity indicated by its traditional use (as a skin sore wash) [10]. Future studies should aim to determine if and which of these PhPs are responsible for the antimicrobial activity observed. This will require a substantial amount of each PhP to be isolated for testing.

## 4. Materials and Methods

### 4.1. General

A Shimadzu liquid chromatography (LC) 2020 series coupled with a diode array detector (DAD) and a mass spectrometer (MS) was used in the analysis of crude extracts and fractions. This system included a reverse-phase semi-preparative column (Phenomenex^®^ (Torrance, CA, USA) Luna Omega 5 µm PS C18 100 × 3.0 mm), and samples were eluted with a binary solvent system consisting of Solvent A (formic acid (0.1%)) and Solvent B (95% acetonitrile, 0.1% formic acid). A volume of 10 μL was injected into the system and eluted from the column (40 °C) at 0.5 mLmin^−1^ with a mobile-phase B held at 5% for 5 min, followed by linear gradients of B from 5 to 95% (5–65 min) where it was held for 10 min. The DAD monitored a UV-visible range between 200 and 600 nm, while the MS with Electron Spray Ionization (ESI) detected a mass to charge (*m*/*z*) range between 50 and 1000 *m*/*z* in dual mode (positive and negative ions). 

A Shimadzu gas chromatography mass spectrometer (GCMS)-QP2010-plus spectrometer was used for sample analysis. Instrument control, acquisition and analysis were conducted using GCSolutions^®^ (Shimadzu, Kyoto, Japan). The analytical column was (ZB-5MS column (30 m)) eluted at 3.5 mLmin^−1^ (column flow), and a total flow of 97.5 mLmin^−1^ injection port was kept at 250 °C, and the detector 250 °C. Temperature of the column was kept at 50 °C for one minute, followed by a 37.5 °C min^−1^ increase until reaching 200 °C, at which point the temperature was kept constant for 10 min. It was increased at 20 °C min^−1^ until reaching 300 °C, where it was held until stop at 30 min. The MS monitored between 40 and 800 *m*/*z* with 0.5 sec event time.

^1^H NMR and ^13^C NMR spectra were recorded with a Bruker Avance 500 spectrometer using a 5 mm SEI probe-Selective Excitation Inverse probe or a Bruker Ascend 500 MHz spectrometer using a 5 mm BBFO probe (broad band plus Fluorine observe probe). Measurements were made in deuterated chloroform (CDCl_3_, with residual CHCl_3_ referenced at 7.26 and 77.00 parts per million (ppm)) with chemical shifts recorded in ppm and coupling constants, *J* values, measured in hertz (Hz).

**Plant collection:** plant specimens were collected at the Greening Australia Site in Thorak Road, Darwin, 12°30′22″ S, 130°57′26″ E, by Greg Leach and deposited at the Northern Territory Herbarium (Voucher ID: Leach 4771).

### 4.2. Plant Extraction and Isolation

*H. coccineum* rhizome was ground to give crude plant material (10.98 g). This material was suspended in methanol (100 mL) and extracted in an ultrasonic bath (30 min). Solvent was removed in vacuo for preliminary analysis. Extract (1.05 g, 9.6 wt/wt%) was resuspended in aqueous methanol (2:1, MeOH:H_2_O, 90 mL) and partitioned with chloroform (3 × 30 mL). Solvents from both fractions were removed under reduced pressure to give aqueous methanol partition (0.80 g, 76 wt/wt%) and a chloroform partition (0.25 g, 23.8 wt/wt%) (Figure 1).


**Solid-Phase Extraction (SPE)**


Some of the aqueous methanol partition (0.5 g) was suspended in methanol (1 mL) and loaded onto a Strata^®^ C-18 SPE (55 μm, 70 Å, Phenomenex^®^, Torrance, CA, USA) and eluted with increasing percentages of aqueous methanol (0%, 30%, 60%, 100% MeOH) in aliquots (200 mL). This procedure produced four fractions (200 mL ea.), which were analysed with LCMS and ^1^H NMR. The 10% MeOH/water fraction contained compounds **1**, **2** and **3,** which were characterised from this mixture. The fraction eluted with 30% MeOH/water contained compound **4**. The fraction eluted with 60% MeOH/water contained compounds **5** and **6**. The fraction eluted with 100% MeOH contained compounds **7** and **8** (Figure 1).


**HPLC of MeOH(aq)**


The aqueous methanol partition of methanolic extract was fractionated with an isocratic method with a flow rate of 1 mLmin^−1^, with mobile-phase B held at 40% until stop at 37.50 min. The column was maintained at a temperature of 45 °C. The UV detector was set to monitor at 217 and 273 nm, and a total of 7 fractions were collected. Previously observed compounds included **1**, **3**, **6**, **10**, and **12**, alongside compound **20**. Across the seven fractions the compounds were present as binary mixtures (Figure 1).


**Column of CHCl_3_**


Crude methanolic extract was partitioned between aqueous methanol and chloroform. The chloroform partition was fractionated over normal-phase silica (Merck (Darmstadt, Germany), 230–400 mesh, 40–63 μm, 60 Å) with the column initially conditioned with 10% ethyl acetate in *n*-hexane. Flash chromatography proceeded with increasing concentrations of ethyl acetate (10–100%) in *n*-hexane, (5.14 L total). A total of 257 fractions (20 mL) were combined into 9 fraction pools based on TLC. The solvent was removed under reduced pressure. Previously observed compounds **4**, **16** and **18** were identified once more, alongside a pure fraction of compound **19** (Figure 1).


**PHWE**


*H. coccineum* (2 × 9.47 g) ground plant material mixed with acid-washed sand (4.32 g) was loaded into a portafilter and extracted with an espresso machine (Breville^®^, Sydney, Australia) (2 × 100 mL). Water was removed via freeze drying, and the extract was resuspended in aqueous methanol (2:1, MeOH:H_2_O, 90 mL). The aqueous methanol solution was extracted with chloroform (3 × 30 mL). LC-DAD-MS analysis of the organic partition was used to identify compounds **7**, **8**, **10** and **12**, based on retention time, UV profile and MS data comparison to compounds isolated and characterized from other fractions. Compounds **9** and **11** were proposed based on mass spectrum data and comparison with the other structures identified in this extract (Figure 1).


**Hydrolysis**


The crude PHWE (16.04 g) was hydrolysed in methanolic (100 mL) HCl (conc. 10 mL) under reflux for one hour. After cooling to room temperature, the hydrolysis products were extracted into chloroform (4 × 30 mL), and both aqueous and organic layers were analysed. The chloroform partition (0.21 g, 1.3 wt/wt%) was fractionated using HPLC, resulting in 12 fractions. Aqueous partition (15.14 g, 94.4 wt/wt%) was filtered through a silica plug to yield α-D-glucose (**2**) (Figure 1).


**HPLC of organic soluble hydrolysis products.**


The fractionation of chloroform extract was obtained with mobile-phase B held at 45% for 5 min followed by linear gradient to 95% of B (5–45 min), where it was held at 95% for 10 min, before a drop to 45% over 2.5 min, where it was held until stop at 60 min. The column was maintained at a temperature of 45 °C, and the UV detector was set to monitor at 273 and 370 nm. A total of 10 fractions were collected, and 6 compounds were identified in these fractions. Binary mixtures of compounds **13**–**14**, **16**–**17**, **17**–**18** were used to characterize them. Compound **15** was the major component of the crude hydrolysis ^1^H NMR, but no ^13^C or 2D NMR data were collected since assignment was consistent with the literature [20] and the compound was not isolated via the fractionation process (Figure 1).

### 4.3. Characterisation

*Sucrose* (**1**): **LCMS *(m*/*z)*** 380.98 [M + H]˙^+^, 378.82 [M − H]˙^−^. **^1^H NMR (300 MHz, CD_3_OD)** δ_H_ 5.39 (d, *J* = 3.8 Hz, 1H, H-1), 4.10 (d, *J* = 8.0 Hz, 1H, H-3′). **^13^C NMR (75 MHz, CD_3_OD)** δ_C_ 93.6 (C1), 79.2 (C3′). NMR data are consistent with the literature [13].

α*-D-glucose* (**2**): **^1^H NMR (300 MHz, CD_3_OD)** δ_H_ 5.11 (d, *J* = 3.7 Hz, 1H, H-1). NMR data are consistent with the literature [13].

β*-D-glucose* (**3**): **^1^H NMR (300 MHz, CD_3_OD)** δ_H_ 4.41 (d, *J* = 7.7 Hz, 1H, H-1), 3.36 (m, 1H, H-3), 3.26 (m, 1H, H-5), 3.22 (d, *J* = 2.3 Hz, 1H, H-4), 3.13 (br s, 1H, H-2). **^13^C NMR (75 MHz, CD_3_OD)** δ_C_ 98.5 (C1), 76.1 (C2), 73.3 (C3), 72.0 (C4), 71.9 (C5). NMR data are consistent with this literature[13].

*6-O-[(6″-O-malonyl)-*β*-D-glucopyranosyl]-5-hydroxy-2-methoxy-7-phenylphenalen-1-one* or *6″-O-malonyldilatrin* (**4**): **UV (λ_max_):** 470, 374, 274 nm. **GCMS (*m*/*z*)** 405.20 [M − Mal]˙^+^, 319.20 [M − Glc − Mal + H]˙^+^. **LCMS *(m*/*z)*** 567.58 [M + H]˙^+^, 565.38 [M − H]˙^−^, 1131.38 [2M − H]˙^−^, 521.38 [M − FA + H]˙^−^. **^1^H NMR (500 MHz, CD_3_OD)** δ_H_ 8.48 (dd, *J* = 7.7, 1.5 Hz, 1H, H-9), 7.58 (s, 1H, H-4), 7.55 (dd, *J* = 7.7, 1.5 Hz, 1H, H-8), 7.48 (dd, *J* = 7.5, 1.7 Hz, 2H, H-3′, H-5′), 7.35 (dd, *J* = 7.5, 1.1 Hz, 1H, H-2′, H-6′), 7.22 (dd, *J* = 4.7, 1.0 Hz, 1H, H-4′), 7.15 (s, 1H, H-3), 4.66 (d, *J* = 7.8 Hz, 1H, H-1″), 4.17 (dd, *J* = 11.8, 5.1 Hz, 1H, H-6″a), 4.00 (dd, *J* = 11.8, 2.0 Hz, 1H, H-6″b), 3.94 (s, 3H, CH_3_O-2), 3.18 (dd, *J* = 9.3, 9.0 Hz, 1H, H-3″), 3.11 (ddd, *J* = 9.0, 5.1, 2.0 Hz, 1H, H-5″), 2.99 (dd, *J* = 9.3, 9.0 Hz, 1H, H-4″), 2.37 (dd, *J* = 9.0, 7.8 Hz, 1H, H-1″). **^13^C NMR (125 MHz, CD_3_OD)** δ_C_ 180.6 (C1), 168.5 (C1‴), 154.0 (C2) 131.1 (C8), 131.0 (C3′, C5′), 127.5 (C6a), 127.2 (C9), 126.7 (C4′), 126.2 (C2′, C6′), 123.6 (C4), 113.8 (C3), 102.2 (C1″), 75.1 (C3″), 72.8 (C2″), 69.3 (C4″), 54.1 (CH_3_O-2). NMR data are consistent with the literature [16].

*6-O-[(6″-O-malonyl)-*β*-D-glucopyranosyl]-2,5-dihydroxy-7-phenylphenalen-1-one* (**5**): **LCMS (*m*/*z*)** 553.45 [M + H]˙^+^, 576.37 [M + Na]˙^+^. **^1^H NMR (500 MHz, CD_3_OD)** δ_H_ 8.46 (d, *J* = 7.6 Hz, 1H, H-9), 7.55 (s, 1H, H-4), 7.53 (d, *J* = 7.6 Hz, 1H, H-8), 7.44–7.28 (m, 5H, H-2′, H-3′, H-4′, H-5′, H-6′), 7.34 (br s, 1H, H-4′), 7.11 (s, 1H, H-3), 4.79 (d, *J* = 7.8 Hz, 1H, H-1″), 3.46 (dd, *J* = 11.6, 5.4 Hz, 1H, H-3″), 3.18 (s, 2H, H-2‴), 2.38 (dd, *J* = 9.4, 7.8 Hz, 1H, H-2″). NMR data are consistent with the literature [15].

*6-O-[(6″-O-malonyl)-*β*-D-glucopyranosyl]-5-hydroxy-7-(4′-hydroxyphenyl)-3H-benzo[de]isochromen-1-one* (**6**): **UV (λ_max_):** 366, 337, 255, 243, 225 nm. **LCMS *(m*/*z)*** 557.01 [M + H]˙^+^, 554.78 [M − H]˙^−^, 511.11 [M − COOH + H]˙^−^. **^1^H NMR (500 MHz, CD_3_OD)** δ_H_ 8.12 (d, *J* = 7.5 Hz, 1H, H-9), 7.40 (d, *J* = 7.5 Hz, 1H, H-8), 7.21 (s, 1H, H-4), 6.80 (d, *J* = 8.3 Hz, 2H, H-3′, H-5′), 5.73 (dd, *J* = 4.2, 1.3 Hz, 1H, H-3), 4.57 (d, *J* = 7.8 Hz, 1H, H-1″), 3.59 (dt, *J* = 11.6, 2.2 Hz, 1H, H-2″), 3.20 (s, 2H, H-2‴), 3.02 (t, *J* = 9.4 Hz, 1H, H-3″), 2.68 (dd, *J* = 9.4, 7.8 Hz, 1H, H-4″). NMR data are consistent with the literature [15].

*Lachnanthocarpone dimethylether* (**7**): **LCMS *(m*/*z)*** 317.11 [M + H]˙^+^, 314.95 [M − H]^˙−^. **^1^H NMR (500 MHz, CD_3_OD)** δ_H_ 8.61 (d, *J* = 8.4 Hz, 1H, H-7), 7.59 (d, *J* = 8.0 Hz, 1H, H-4), 7.55 (d, *J* = 8.4 Hz, 1H, H-8), 7.42 (d, *J* = 8.0 Hz, 2H, H-3′, H-5′), 7.38 (t, *J* = 8.0 Hz, 1H, H4′), 7.34 (d, *J* = 1.7, 2H, Hz H-2′, H-6′), 6.88 (d, *J* = 8.0 Hz, 1H, H-5), 6.81 (s, 1H, H-3), 4.07 (s, 3H, CH_3_O-2), 3.84 (s, 3H, CH_3_O-6). **^13^C NMR (125 MHz, CD_3_OD)** δ_C_ 179.9 (C1), 130.6 (C8), 129.3 (C4′), 128.7 (C3′, C5′), 128.2 (C7), 127.7 (C2′, C6′), 111.8 (C3), 104.5 (C5), 55.7 (CH_3_O-6), 55.2 (CH_3_O-2). NMR data are consistent with the literature [22].

*Haemodordioxolane* (**8**): **UV (λ_max_):** 513, 374, 280, 218 nm. **LCMS *(m*/*z)*** 319.18 [M + H]˙^+^. **^1^H NMR (500 MHz, CD_3_OD)** δ_H_ 8.54 (d, *J* = 7.5 Hz, 1H, H-9), 8.26 (s, 1H, H-4), 7.57 (d, *J* = 7.5 Hz, 1H, H-8), 7.48 (d, *J* = 1.6 Hz, 2H, H-3′, H-5′), 7.43 (d, *J* = 1.7 Hz, 2H, H-2′, H-6′), 6.09 (s, 1H, H-10). NMR data are consistent with the literature [7].

6-(β-D-glycopyranosyl)-5-hydroxy-7-phenyl-1H-benzo[*de*]isochromen-1-one (**9**): **UV (λmax):** 366, 334, 260, 220 nm. **LCMS *(m*/*z)*** 455.14 [M + H]˙^+^, 452.96 [M − H]˙^−^, 293.12 [M-Glc]˙^+^.

*6-(β-D-glycopyranosyl)-5-hydroxy-7(4′-hydroxyphenyl)-1H,3H-benzo[de]isochromen-1-one* (**10**): **UV (λ_max_):** 385, 326, 255, 223 nm. **LCMS *(m*/*z)*** 939.09 [2M − H]^˙−^, 471.10 [M + H]˙^+^, 469.04 [M − H]˙^−^, 309.10 [M-Glc]˙^+^. **^1^H NMR (500 MHz, CD_3_OD)** δ_H_ 8.14 (d, *J* = 7.5 Hz, 1H, H-9), 7.42 (d, *J* = 7.5 Hz, 1H, H-8), 7.35–7.33 (m, 2H, H-2′, H-6′), 7.23 (s, 1H, H-4), 6.96–6.94 (m, 2H, H-3′, H-5′), 5.75 (dd, *J* = 15.0, 1.2 Hz, 1H, H-3a), 5.72 (dd, *J* = 15.0, 1.2 Hz, 1H, H-3b), 4.67 (d, *J* = 8.0 Hz, 1H, H-1″), 3.59 (dd, *J* = 11.2, 2.0 Hz, 1H, H-6″a), 3.46 (dd, *J* = 11.2, 5.7 Hz, 1H, H-6″b), 3.05–3.02 (m, 1H, H-4″), 3.19 (dd, *J* = 10.0, 8.7 Hz, 1H, H-3″), 2.92 (ddd, *J* = 9.8, 5.6, 2.0 Hz, 1H, H-5″), 2.66 (dd, *J* = 9.3, 7.8 Hz, 1H, H-2″). **^13^C NMR (125 MHz, CD_3_OD)** δ_C_ 167.6 (C1), 146.5 (C7), 139.6 (6), 131.8 (C8), 130.4 (C2′, C6′), 128.9 (C9a), 127.1 (C9), 126.7 (C3a), 126.6 (C9b), 114.8 (C3′, C5′), 116.7 (C4), 104.8 (C1″), 78.1 (C5″), 77.8 (C3″), 74.8 (C2″), 71.5 (C4″), 70.7 (C3), 62.9 (C6″). NMR data are consistent with the literature [14].

6*-(β-D-glucopyranosyloxy)-5-hydroxy-2-methoxy-7-phenyl-1H-phenalen-1-one* (**11**): **UV (λmax):** 470, 374, 274, 244 nm. **LCMS *(m*/*z)*** 1131 [2M − H]^˙−^, 521 [M − H]˙^−^, 567 [M + FA]^˙+^, 319 [M − Glc − Ac + H]˙^+^.

*Dilatrin* (**12**): **UV (λ_max_):** 473, 374, 275, 232 nm. **LCMS *(m*/*z)*** 959.12 [2M − H]^˙−^, 481.10 [M + H]˙^+^, 479.09 [M − H]˙^−^, 319.09 [M − Glc]˙^+^. **^1^H NMR (500 MHz, CD_3_OD)** δ_H_ 8.39 (d, *J* = 7.6 Hz, 1H, H-8), 7.48 (s, 1H, H-4), 7.47 (d, *J* = 7.6 Hz, 1H, H-9), 7.36–7.19 (m, 5H, C2′, C3′, C4′, C5′, C6′), 7.13 (s, 1H, H-3), 4.71 (d, *J* = 7.8 Hz, 1H, H-1″), 3.86 (s, 3H, CH_3_O-2), 3.51 (dd, *J* = 12.0, 2.3 Hz, 1H, H-6″a), 3.38 (dd, *J* = 12.0, 5.4 Hz, 1H, H-6″b), 3.11 (t, *J* = 9.0 Hz, 1H, H-3″), 2.89 (q, *J* = 9.0 Hz, 1H, H-4″), 2.85 (ddd, *J* = 9.0, 5.4, 2.3 Hz, 1H, H-5″), 2.29 (dd, *J* = 9.0, 7.8 Hz, 1H, H-2″). **^13^C NMR (125 MHz, CD_3_OD)** δ_C_ 181.5 (C6), 153.6 (C5), 149.1 (C9), 148.7 (C2), 145.4 (C1′), 141.6 (C1), 132.1 (C8), 129.2 (C6a), 128.6 (C7), 128.3 (C9a), 125.1 (C4′), 124.7 (C3′, C5′), 124.3 (C3a), 122.4 (C3), 121.5 (C2′), 121.4 (C6′), 121.3 (C9b), 116.4 (C4), 104.6 (C1″), 77.9 (C5″), 77.6 (C3″), 74.6 (C2″), 70.9 (C4″), 62.6 (C6″), 56.6 (CH_3_O-2). NMR data are consistent with the literature [25].

*Haemodorone* (**13**): **LCMS *(m*/*z)*** 318.86 [M − H]^˙−^. **^1^H NMR (500 MHz, CDCl_3_)** δ_H_ 8.41 (d, *J* = 8.2 Hz, 1H, H-9), 8.29 (s, 1H, H-4), 7.55 (d, *J* = 8.2 Hz, 1H, H-8), 7.44–7.41 (m, 2H, H-2′, H-6′), 7.39–7.36 (m, 3H, H-3′, H-4′, H-5′), 3.68 (s, 3H, CH_3_O-6). **^13^C NMR (125 MHz, CDCl_3_)** δ_C_ 135.6 (C8), 131.8 (C9), 130.2 (C3′, C4′, C5′), 129.4 (C2′, C6′), 126.9 (C4), 56.8 (CH_3_O-6). NMR data are consistent with the literature [6].

*2-hydroxy-6-methoxy-9-phenyl-1H-phenalen-1-one* (**14**): **HRMS** (C_20_H_14_O_3_), calc’d 303.1016, obs’d 303.0648 [M + H] ^+^, 335.0908 [M + CH_3_OH + H] ^+^. **^1^H NMR (500 MHz, CD_3_OD)** δ_H_ 8.71 (d, *J* = 8.4 Hz, 1H, H-7), 7.63 (d, *J* = 8.1 Hz, 1H, H-4), 7.56 (d, *J* = 8.4 Hz, 1H, H-8), 7.46–7.41 (m, 5H, H-2′, 3′, 4′, 5′, 6′), 7.07 (s, 1H, H-3), 6.90 (d, *J* = 8.1 Hz, 1H, 5), 4.01 (s, CH_3_O-6). **^13^C NMR (125 MHz, CD_3_OD)** δ_C_ 133.32 (C4), 122.76 (C3). NMR data are consistent with the literature [20].

*Xiphidone* (**15**): **UV (λ_max_):** 562, 401, 266, 219 nm. **LCMS *(m*/*z)*** 333.20 [M + H]^˙+^, 331.19 [M − H]^˙−^. **^1^H NMR (500 MHz, CDCl_3_)** δ_H_ 10.12 (s, 1H, OH-6), 8.71 (d, *J* = 8.2 Hz, 1H, H-9), 7.56 (d, *J* = 8.2 Hz, 1H, H-8), 7.49 (s, 1H, H-4), 7.44–7.36 (m, 5H, H-2′, 3′, 4′, 5′, 6′), 7.10 (s, 1H, H-3), 3.98 (s, 3H, H-11), 3.75 (s, 3H, H-10). NMR data are consistent with the literature [20]. 

*2,5-dimethoxy-1*H*-naphtho [2,1,8-mna]xanthen-1-one* (**16**): **LCMS *(m*/*z)*** 331.20 [M + H]˙^+^. **^1^H NMR (500 MHz, CD_3_OD)** δ_H_ 8.81 (d, *J* = 8.0 Hz, 1H, H-12), 8.10 (d, *J* = 8.0 Hz, 1H, H-11), 7.88 (br d, *J* = 7.5 Hz, 1H, H-10), 7.65 (s, 1H, H-4), 7.55 (dd, *J* = 8.5, 1.5 Hz, 1H, H-8), 7.44 (dd, *J* = 8.5, 1.5 Hz, 1H, H-7), 7.38 (ddd, *J* = 8.5, 7.5, 1.5 Hz, 1H, H-9), 7.05 (s, 1H, H-3), 4.09 (s, 3H, CH_3_O-5), 4.01 (s, 3H, CH_3_O-2). **^13^C NMR (125 MHz, CD_3_OD)** δ_C_ 131.4 (C12), 120.3 (C4), 117.4 (C7), 116.2 (C11), 111.9 (C3), 56.4 (CH_3_O-5), 55.7 (CH_3_O-2). NMR data is consistent with literature [21].

*Haemodordiol* (**17**): **LCMS *(m*/*z)*** 304.97 [M + H]^˙+^, 303.64 [M − H]^˙−^. **^1^H NMR (500 MHz, CDCl_3_)** δ_H_ 8.61 (d, *J* = 8.9 Hz, 1H, H-9), 7.76 (d, *J* = 8.9 Hz, 1H, H-8), 6.53 (s, 1H, H-3), 4.01 (s, 3H, CH_3_O-3). **^13^C NMR (125 MHz, CDCl3)** δ_C_ 146.9 (C8), 131.4 (C9), 56.9 (CH_3_O-3). NMR data are consistent with the literature [8].

*5-hydroxy-2-methoxy-6-oxabenzo[def]crysen-1-one or 5-hydroxy-2-methoxy-1H-naphtho [2,1,8-mna]xanthen-1-one* (**18**): **LCMS *(m*/*z)*** 317 [M + H]˙^+^, 315 [M − H]^˙−^. **^1^H NMR (500 MHz, CD_3_OD)** δ_H_ 8.52 (d, *J* = 7.9 Hz, 1H, H-12), 8.39 (dt, *J* = 8.2, 2.1 Hz, 1H, H-10), 8.32 (d, *J* = 8.0 Hz, 1H, H-11), 7.79 (s, 1H, H-4), 7.65 (d, *J* = 7.9 Hz, 1H, H-8), 7.49 (d, *J* = 7.9 Hz, 1H, H-7), 7.39 (s, 1H, H-3), 3.85 (s, 3H, CH_3_O-2). **^13^C NMR (125 MHz, CD_3_OD)** δ_C_ 132.3 (C8), 130.0 (C12), 124.6 (C10), 122.3 (C4), 117.9 (C7), 116.3 (C11), 112.8 (C3). NMR data is consistent with the literature. 24 **^1^H NMR (500 MHz, CDCl_3_)** δ_H_ 8.84 (d, *J* = 8.0 Hz, 1H, H-12), 8.22 (d, *J* = 8.0 Hz, 1H, H-10), 8.13 (d, *J* = 8.0 Hz, 1H, H-11), 7.68 (s, 1H, H-4), 7.58 (dd, *J* = 7.6, 6.9 Hz, 2H, H-7, H-8), 7.43–7.40 (m, 1H, H-9), 4.02 (s, 3H, CH_3_O-2). NMR data are consistent with the literature [8].

*6-O-[6″-O-malonyl-β-D-glucopyranosyl]-5-hydroxy-7-phenyl-3H-benzo[de]isochromen-1-one* (**19**): **LCMS *(m*/*z)*** 540.98 [M + H]˙^+^, 563.63 [M + Na]˙^+^. **^1^H NMR (500 MHz, CD_3_OD)** δ_H_ 8.13 (d, *J* = 7.5 Hz, 1H, H-9), 7.41 (d, *J* = 7.5 Hz, 1H, H-8), 7.38 (br s, 1H, H-4′), 7.33 (br s, 2H, H-2′, H-6′), 7.22 (s, 1H, H-4), 5.76 (dd, *J* = 3.5, 1.3 Hz, 1H, H-3), 4.61 (d, *J* = 7.4 Hz, 1H, H-1″), 3.60 (dd, *J* = 12.0, 2.8 Hz, 1H, H-6″b), 3.46 (dd, *J* = 12.0, 5.6 Hz, 1H, H-6″a), 3.19 (dd, *J* = 9.7, 8.9 Hz, 1H, H-3″), 3.04–3.00 (m, 1H, H-5″), 2.94–2.89 (m, 1H, H-4″), 2.67 (ddd, *J* = 9.7, 5.6, 2.6 Hz,1H, H-2″), 2.66 (s, 2H, H-2‴). **^13^C NMR (125 MHz, CD_3_OD)** δ_C_ 165.9 (C1), 145.1 (C7), 144.8 (C1′), 137.3 (C6), 131.6 (C8), 129.2 (C4′), 127.4 (C9a), 127.3 (C9), 126.4 (C2′, C6′), 124.7 (C9b), 118.1 (C6a), 116.6 (C4), 104.7 (C1″), 78.0 (C5″), 76.6 (C3″), 75.2 (C2″), 71.3 (C4″), 70.6 (C3), 63.0 (C6″), 40.6 (C2‴). NMR data are consistent with the literature [15].

*Fructose* (**20**): **^1^H NMR (500 MHz, CD_3_OD)** δ_H_ 4.03 (d, *J* = 8.2 Hz, 1H, H-5), 3.56 (m, 2H, H-6). **^13^C NMR (125 MHz, CD_3_OD)** δ_C_ 79.4 (C5), 64.3 (C6). NMR data are consistent with the literature [13].

### 4.4. Soil Soxhlet Extraction

A portion of soil (11.42 g), which surrounds the rhizome, was placed in a thimble and extracted via soxhlet using methanol (100 mL, 11H 15 min). The solvent was removed under reduced pressure. The sample was analysed using LCMS and GCMS. **GCMS**: glycerol (6.71 min), ethyl pyruvate (7.70 min), D-hexose (11.42 min).

### 4.5. Antimicrobial Assay

Antimicrobial testing protocols for all microbes excluding H. influenzae have been previously reported [26] and were based on Clinical Laboratory Standards Institute (CLSI) for antimicrobial susceptibility testing [27,28]. In brief, the extract was tested against bacterial reference strains *Escherichia coli* (ATCC 25922) and *Pseudomonas aeruginosa* (ATCC 25668), *Staphylococcus aureus* (ATCC 25923), *Streptococcus pyogenes* (ATCC 19615) and yeast reference strain *Candida albicans* (ATCC 90029). The extract was tested at a single concentration of 20 mg/mL in 2% DMSO in a final volume of 150 μL. Microbe growth was assessed via optical density at 595 nm using a plate reader (Victor X2, Perkin Elmer). Antimicrobial activity was defined as an 80% or greater reduction in optical density at 20 h for bacteria, and 24 h for yeast, compared to the relevant microbe growth control.

The antimicrobial testing protocols for *H. influenzae* were adapted from the British Pharmacopeia’s Appendix XIV P *Determination of Bactericidal, Fungicidal or Yeasticidal Activity of Antiseptic Medicinal Products* [29]. An overnight culture of reference strain *H. influenzae* (ATCC 49274) grown on chocolate agar plates was used to make a suspension of approximately 5 × 10^7^ CFU/mL in sterile saline (0.9%), and was added to an equal volume of the crude extract at a final concentration of 20 mg/mL in a microlitre plate in a final volume of 100 µL. The plate was incubated at 33 degrees at 180 rpm for 10 min; then the bacteria extract was diluted by a factor of 10^4^, and 100 µL was plated on chocolate agar plates, which were incubated overnight at 37 degrees. The colonies were counted and the percentage inhibition was calculated compared to the microbe only control. The results are from a minimum of three independent experiments.

## Figures and Tables

**Figure 1 molecules-28-07422-f001:**
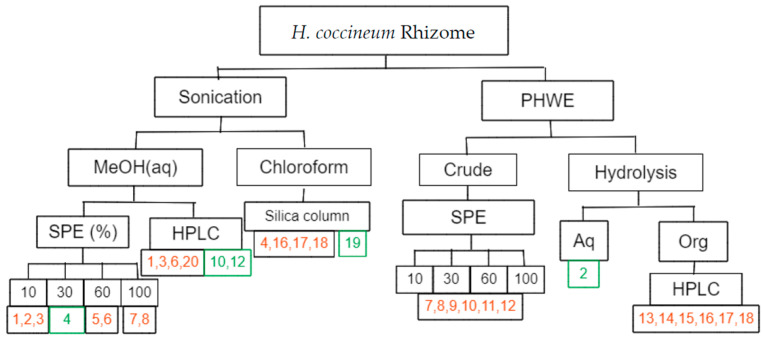
Compound isolation flowchart. The numbers in black represent the percentage of methanol (aq) used to elute that SPE fraction. Compound numbers in orange indicate compounds characterised from sample mixtures. Compounds in green were characterised as single compounds.

**Figure 2 molecules-28-07422-f002:**
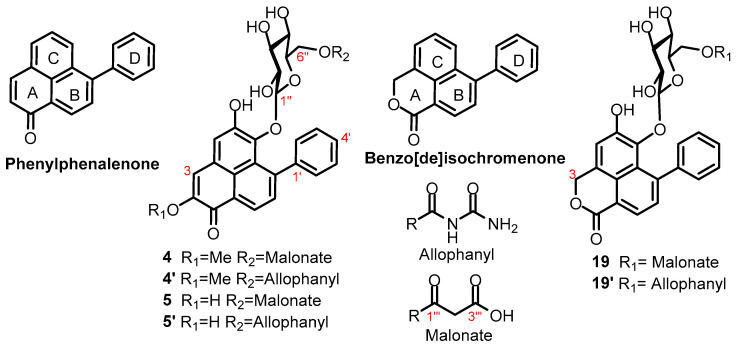
Structures of phenylphenalenone, benzo[*de*]isochromenone, allophanyl, malonate and compounds **4**, **4′**, **5**, **5′**, **19** and **19′**. (Please refer to Appendix A for clarification on naming and numbering of structures).

**Figure 3 molecules-28-07422-f003:**
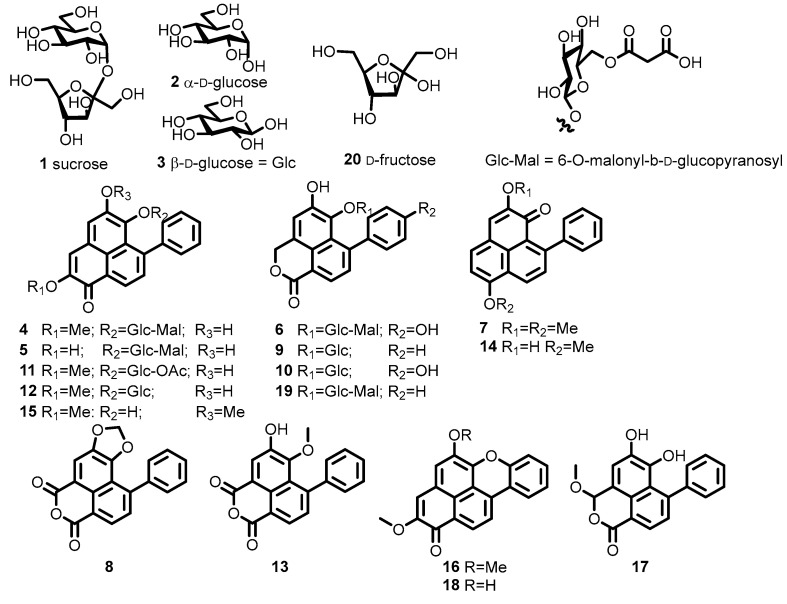
Structures of all compounds isolated from the rhizome of *H. coccineum*.

**Table 1 molecules-28-07422-t001:** Accurate mass predictions calculated for allophanyl and malonate subunits.

	Formula	Accurate Mass
Allophanyl	C_2_H_4_N_2_O_3_	104.022193
Malonate	C_3_H_4_O_4_	104.010960

**Table 2 molecules-28-07422-t002:** Carbonyl chemical shift reported by Opitz [16] and the difference to the predicted allophanyl shift (red) and malonate shift (blue).

** 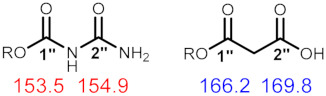 **						
**Compound**	1″	Δ_urea_	Δ_mal_	2″	Δ_urea_	Δ_mal_
**4′**	168.2	14.7	2.0	169.9	15.0	0.1
**5′**	169.3	15.8	3.1	171.5	16.6	1.7
**19′’**	170.0	16.5	3.8	172.4	17.5	2.6

**Table 3 molecules-28-07422-t003:** HREMSIMS data for compounds **4′**, **5′** and **19′**.

	Formula	Accurate Mass	δ
Reported	Compound **4′** [M + H]	567.1486	-
Allophanyl	C_28_H_27_N_2_O_11_	567.1615	0.0129
Malonate	C_29_H_27_O_12_	567.1503	0.0017
Reported	Compound **5′** [M + H]	553.1278	-
Allophanyl	C_27_H_25_N_2_O_11_	553.1458	0.0180
Malonate	C_28_H_25_O_12_	553.1346	0.0068
Reported	Compound **19′** [M + H]	541.1378	-
Allophanyl	C_26_H_25_N_2_O_11_	541.1458	0.0080
Malonate	C_27_H_25_O_12_	541.1346	0.0032

## Data Availability

Not applicable.

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
