# Peer review of "Differentiating Dyes: A Spectroscopic Investigation into the Composition of Scarlet Bloodroot (Haemodorum coccineum R.Br.) Rhizome"

_molecules, 2023, doi:10.3390/molecules28217422_

Round 1

Reviewer 1 Report

Comments and Suggestions for Authors

Dear authors

I checked your manuscript and the content was intriguing. However, there are some comments that should be addressed before publishing this paper. Please check the following comments and amend the manuscript accordingly. 

1- Figure F2 has lots of noises. How the respected authors optimized the observed noises in this chromatogram?

2- The respected authors only isolated the mentioned compounds from the studied plant. It is better to couple at least one or two biological assays (such as toxicity assay) with the current data to enhance the quality of outcomes. 

3- The authors only isolated chemical compounds of the studied plant rhizome. To calculate the frequency and distribution of the identified compounds in this plant, I highly recommend the authors to survey other parts of this plant and to know which compounds are abound in other tissues of this plant. This helps you to report comprehensive analytical results for identification of the isolated compounds. 

4- Please prepare a flowchart for the M&M section of this paper. 

Comments on the Quality of English Language

English is fine but minor changes should be considered during revision of this manuscript

Author Response

Reviewer comments addressed in blue, underneath the respective comment.

Reviewer 2 Report

Comments and Suggestions for Authors

The work has significant value in dye industry and also in medical science. However, the manuscript need to revise vigorously. The position of full-stops needs to be checked throughout the manuscript.

line 20-23: aim and method will be more suitable after background sentences within the abstract (i.e., in line 16). Mentioning the funding sources is not mandatory in the abstract section.

line 25: Haemodorum coccineum--- will be in italicized form.

line 42-44: As earlier works on Haemodorum corymbosum (a synonym of H. coccineum) were done and published, rewrite the statements. Also applicable throughout the manuscript. These two names are given for a single plant species. Therefore, this study can not be treated as first-time work on this plant.

Result & Discussion: Firstly, write the results and then discuss them. Otherwise, finding the results would be difficult for a reader.

line 276-284: rewrite compactly.

Author Response

(The authors gave the same response as above.)

Reviewer 3 Report

Comments and Suggestions for Authors

The manuscript ID molecules-2666372 describes the chemical characterization of pigments produced by Haemodorum coccineum rhizome. Although this manuscript has some interesting elements, novelty-related concerns limit further consideration.

Major points

1.      The manuscript lacks novelty and merit since the isolated compounds are known, and no isolated compound is new. The above is not good because such explanations about structural elucidation do not make much sense because they are known compounds. These detailed spectral and structural descriptions are often made for unprecedented compounds with unreported structures. Therefore, all these descriptions and spectral data in the M&M section are unnecessary and should be removed. If this is done, the manuscript, despite the notorious laboratory work behind it, turns out to be very simple and without apparent novelty. In other words, a different source of known compounds does not justify novelty, and a large amount of known compounds does not justify scientific merit.  In this sense, I recommend that the authors add some elements of novelty and merit, such as a biological and/or chemical activity for these isolated pigments. Otherwise, I regret to insist that the manuscript has no novelty and merit from the chemical point of view.

2.      Based on the previous comment, the manuscript is highly descriptive and lacks an adequate discussion.

3.      Conclusions summarize results, and conceptual findings from a mechanistic point of view are missing.

Minor points

1.      Line 16: The plat part (e.g., rhizome) should be informed in the abstract.

2.      Line 17: The type of spectral characterization (NMR, MS, IR, or all) should be informed in the abstract.

3.      Line 23: A conclusive sentence is missing in the abstract.

4.      Line 24: According to the journal format, this graphical abstract-like figure is not required here.

5.      Line 25: Haemodorum coccineum  in italics. Be consistent throughout the manuscript.

6.      Line 49-53: The aim and scope of this study is unclear. The authors should improve this passage to give readers a better aim and scope.

7.      Line 80: There are three number colors (green, orange, and gray), but only two are defined. The authors should revise the accuracy of this information.

8.      All structural elucidation descriptions and explanations are unnecessary and should be removed because the isolated compounds are known.

9.      The NMR spectra are included for only two compounds (12 and 19). Why were the spectra for the other sixteen isolated compounds not included? In addition, the MS spectra plots are missing.

10.   The spectral data of known compounds in the M&M section are unnecessary.

Comments on the Quality of English Language

Detailed scrutiny should be performed throughout the manuscript to revise some grammar and stylistic issues.

Author Response

(The authors gave the same response as above.)

Round 2

Reviewer 2 Report

Comments and Suggestions for Authors

The revised version of the manuscript has improved and may be suitable for publication after minor revision. Yet I am not very satisfied with the the Results and Discussion section. Yet it can be improved to an outstanding level. Readers can not easily find out the key results of the article; they need to invest much time and maybe bored. Further, a few specific comments are given below: 

Line 16: Haemodoracaea--- check the term

line 29-30: Keywords--- arrange alphabetically.

line 73-79: Authors avoided mentioning the name of some compounds like 5, 6, 10, etc.--- Are these less important or less quantity compounds?

Reviewer 3 Report

Comments and Suggestions for Authors

The authors addressed the minor points, and the manuscript improved on that. However, the major concerns were globally explained and argued with tangential statements, so they were not convincing enough. In this regard, I insist on the fact that the R&D section is very extensive and descriptive of known compounds. The authors added a passage regarding very preliminary antimicrobial activity, but this paragraph and results are laconically described and discussed. In addition, no outcome table or other results format is provided, and the isolated compounds were not evaluated but only extracts. These facts limit its further consideration.
